# Effects of Shortening Replacement with Extra Virgin Olive Oil on the Physical–Chemical–Sensory Properties of Italian Cantuccini Biscuits

**DOI:** 10.3390/foods11030299

**Published:** 2022-01-23

**Authors:** Angelo Maria Giuffrè, Manuela Caracciolo, Marco Capocasale, Clotilde Zappia, Marco Poiana

**Affiliations:** Dipartimento di Agraria, Università degli Studi Mediterranea di Reggio Calabria, 89124 Reggio Calabria, Italy; manuela.caracciolo@unirc.it (M.C.); marcocapocasale@outlook.it (M.C.); czappia@outlook.it (C.Z.); mpoiana@unirc.it (M.P.)

**Keywords:** bakery product, biscuit, Cantuccini, cookie, fat, lipid oxidation, shelf-life

## Abstract

Olive oil is recognised for its beneficial effects on human health, mainly due to it containing oleic acid (a monounsaturated fatty acid), whereas fats of animal origin or margarine, which are often used in recipes for biscuit production, contain mainly saturated fatty acids. The aim of this study was to evaluate the shelf-life and physicochemical properties of biscuits and of the fats contained in original recipe Italian Cantuccini biscuits (50% cow’s butter and 50% margarine). Additionally, the sensory properties of the biscuits were evaluated, including their colour, appearance, taste, flavour, texture and overall acceptability. At the same time, the fat composition of the original recipe was also modified to contain 30% cow’s butter and 70% extra virgin olive oil, in order to replace an aliquot of the saturated fatty acid content with unsaturated fatty acids, in particular with one monounsaturated fatty acid, oleic acid. Colour (CIELab), water activity, relative humidity, hardness and fracturability analyses were conducted on Cantuccini biscuits. Colour (CIELab), free acidity, spectrophotometric characteristics, DPPH assay and fatty acid methyl ester (FAMEs) analyses were conducted on the fat extracted from Cantuccini biscuits prepared from both the original and modified recipes.

## 1. Introduction

The global biscuits market is segmented by category into sweet biscuits and crackers and savoury biscuits. The global market was evaluated at USD 106.232 billion in 2020, and it is estimated to reach a compound annual growth rate of 4.97% during the period 2021–2026 [1]. Sweet biscuits include plain biscuits, cookies, sandwich biscuits, chocolate-coated biscuits and other sweet biscuits. Each country and region produces its own types of sweet biscuits, with their differences depending on the local availability of the ingredients used in their recipes.

Cantuccini are traditional sweet biscuits originally from Tuscany (Italy) but now found throughout Italy. They are dry biscuits that are filled with almonds, which are made by cutting slices of dough while still hot, before being the dough slices to the oven for a few minutes to reach their final consistency.

Cantuccini have a European Protected Geographical Indication (PGI) [2], which is attributed to products for which at least one of the stages of production, processing or preparation takes place in the region. The Italian product specification [3] includes the following requirements: the use of almonds (at least 17%), crystalline or granulated sugar (20–40%) and cow’s butter (at quantities greater than 1.5%). However, if the producer does not apply the PGI certification to their Cantuccini, the recipe can be modified.

Lipids in bakery products play a fundamental role from both technical and edible point of views. Lipids used in the bakery industry are oils or fats, which are respectively liquid or solid or semi-solid at room temperature. The differences in physical consistency are due to the triacylglycerol composition and in particular to the fatty acids and their melting points. Saturated fatty acids are solid or semi-solid at room temperature, whereas unsaturated fatty acids are liquid. The shorter the chain length and the lower the number of double bonds, the lower the melting point. Fats are easier to transport and less susceptible to oxidation than oils. At the same time, saturated fatty acids (SFAs), which are widely present in fats, and cholesterol (widely present in animal fats) are considered responsible for cardiovascular diseases [4]. The lipid content in biscuits usually ranges from 10 to 20% of the total weight, and consequently a daily portion of 50 g (i.e., 3–5 biscuits) contains 5–10 g of lipids. In light of the above, it is essential to reduce the amounts of saturated fats and animal fats in the human diet. In addition, it is well known that nowadays there is a high demand for high nutritional value foods. Extra virgin olive oil (EVOO) contains less than 0.5% cholesterol out of the total sterol content [5], 14–22% SFAs out of the total fatty acid content [6,7], a good oleic/linoleic acid ratio [8] and antioxidants such as phenols and tocopherols [9]. EVOO is one of the most common foods in the Mediterranean diet, and in 2018 Calabria was the second Italian region for olive oil production (46,964 tons) out of the total Italian production of 264,101 tons [10].

From an edible point of view, the fat in biscuits is important for its nutritional value, for its tendency for oxidation and for the consequential radical formation. From a technical point of view, when fats are mixed with flour that has not yet been hydrated, particles of fats wrap around the particles of flour, reducing the dough’s elasticity [11]. High elasticity is not advisable in biscuit dough because it causes the dough to shrink after lamination [12]. In studies on beaten pastry dough, fat was found to influence both the stability and size of air bubbles before baking and at initial baking stages [13,14,15]. In the initial baking process, fat prevents the coalescence of bubbles that are destroyed with the increase in baking temperature. The longer the bubbles remain in a dough, the larger the final volume of the baked products [16]. Fats can help leaven a product due to the incorporation of air [17].

In addition, fat has a lubricating action, which prevents the product from sticking to baking trays and causes the melting sensation when a biscuit is eaten.

To date in the literature, there have been no studies on the physical–chemical–sensory evolution of Cantuccini during storage. Scarce information exists on biscuits [18] and cookies [19] incorporated with EVOO as a total or partial shortening substitution.

The aim of this paper was to study the shelf-life of Cantuccini biscuits produced using the original recipe containing only cow’s butter and margarine, as well as the effects of partial shortening replacement with extra virgin olive oil over one year’s storage.

## 2. Materials and Methods

### 2.1. Cantuccini Packaging

Each package contained 35 biscuits weighing 200 g in total. For this experiment, biscuits were made using two recipes, which differed only in the type of fats or oils used. In the first (original) recipe, 7% fat was used (out of the total ingredients), at a ratio of 50% cow’s butter to 50% margarine. In the second (modified) recipe, 11% fat was used (out of the total ingredients), at a ratio of 30% cow’s butter to 70% extra virgin olive oil. This modified recipe was applied after testing the consistency and pleasantness of the Cantuccini. This ratio was applied so as not to distort the type of product and to provide the right compromise between tradition and innovation. The shortening reduction led to the need for an increase in the EVOO to maintain the original characteristics of the biscuits as much as possible. The technical data on the packaging were: duplex laminate; width: 410 mm; cut off: 285 mm; thickness: 74 μm; material identification code: PP 5; core diameter: 76 mm; outer diameter: 280 mm; unit weight: 68.9 g/m^2^; weight per pack: 8.05 g; heat sealing: both sides; oxygen barrier: very low >150; water vapour barrier: high (0.50–3.00); thermal treatment: none; composition: outer layer: COEXPP 30 μm; inner layer i.e., side in contact: COEXPP BL MAS 40 μm.

### 2.2. Cantuccini Preparation

The original recipe (OR) for Cantuccini biscuits contained 50% cow’s butter and 50% margarine. The modified recipe (MR) contained 30% cow’s butter and 70% extra virgin olive oil. Sugar, margarine, butter and honey were placed in a mixer. The ingredients were mixed until smooth. At this point the eggs, baking powder and flour (00 type) were added. Finally, when the mixture was homogeneous, the whole almonds were added. The dough was kneaded until it was homogeneous, without lumps and of a good consistency. The dough was left for twenty minutes. At this point, pieces of dough weighing 2 kg were passed under the leveler before being cut into small pieces, which were placed in the baking pans and baked at 200 °C for 20 min. The Cantuccini were removed from the oven and cooled to room temperature before packaging. 

### 2.3. Biscuits Analyses

#### 2.3.1. Texture Analysis

The hardness and fracturability of Cantuccini were determined using a texture analyzer (TA-XT2 Texture Analyzer, Stable Microsystems, Surrey, UK) equipped with a 25 kg load cell and a three-point bending rig (HDP/3PB). The parameters of hardness and fracturability were calculated as absolute values.

#### 2.3.2. Colour—Instrumental Determination

Colour was determined using a colourimeter (Konica Minolta, model CM-A177) measuring *L**, Chroma *a** (*a**) and Chroma *b** (*b**). If a colour is expressed in CIELAB, *L** denotes lightness, *a** defines the red/green value and *b** the yellow/blue value. In this experiment, colour determination was conducted on both the external (the top of the biscuit) and internal sides (the side of the cut) of Cantuccini biscuits.

#### 2.3.3. Water Activity (aw)

Aw is the ratio between the vapour pressure of a food system when it is not influenced by the environment (Pv) and the saturation vapour pressure of distilled water (Pa) in the same conditions [20]. The aw determination was made at 20 °C by means of an Aqualab Model series 3TE Decagon Devices, Inc. (Pullman, Washington, WA, USA).

#### 2.3.4. Relative Humidity (RH)

The RH (moisture content) was determined according to the method proposed by [21], with heating at 103 °C ± 2 °C until constant weight.

#### 2.3.5. Sensory Analysis

A 20-member sensory panel was involved (ten men and ten women aged 19 to 60 years). The panelists were trained to consider six attributes, namely colour, appearance, taste, flavour, texture and overall acceptability. Each attribute was evaluated on a 10-point hedonic scale (1–10) ranging from 10 (like extremely) to 1 (dislike extremely) for each characteristic and with 5 representing minimum quality. The test was conducted in a sensory laboratory with individual booths and Cantuccini biscuits were given in white plates in a room prepared as suggested by ‘Sensory Analysis—General Guidance for the Design of Test Rooms’ [22]. The panelists conducted their evaluation in triplicate with a 2 day time lapse at 24 °C room temperature, with samples tempered for 60 min. No odour and minimal noise were perceived by the panelists. The colour of the walls and furnishings in the testing area was white. Only one assessor was present, managing a group of five panelists in a 30 min session.

Colour expressed the degree of toasting. Appearance defined the shape of the biscuit. Taste and flavour defined the freshness of Cantuccini. Texture defined the crunchiness and overall acceptability was the sum of the previous five attributes. The sensory analysis was conducted prior to inform the panelists with regard to the Cantuccini composition.

### 2.4. Analysis of Fat

#### 2.4.1. Fat Extraction

Firstly, the almonds were separated from the biscuits and then reduced to a powder. Secondly, the lipid fraction was extracted from the powder of biscuits using a Soxhlet apparatus with *n*-hexane as the extracting solvent. Lastly, the solvent was evaporated using a Rotavapour apparatus under vacuum at 25 °C and fat was stored until analyses [23].

#### 2.4.2. Determination of Colour

A Minolta Chroma Meter CR-400 instrument and a Minolta transparent (base and side) special glass container (5.0 cm, 6.0 cm high) with a cylindrical shape was used. One centimetre of fat was added to the glass container and the colour was determined. A further 1 cm was added and the colour was determined again. These two measurements were conducted in triplicate. The CIELab scale was used: *L** (brigthness) ranges between 0 (black) and 100 (white); *a** ranges between −90 (green) and +90 (red); *b** ranges between −90 (blue) and +90 (yellow). The chroma was calculated using the formula discussed by Pathare et al. [24].

#### 2.4.3. Free Acidity (FA)

A 1 g aliquot of the lipidic fraction was dissolved in a 50 mL ethanol/diethyl ether (1:1), after which titration was conducted with a 0.1 N NaOH solution. The results were expressed as g oleic acid/100 g (Consleg, Annex II) [5].

#### 2.4.4. Spectrophotometric Indexes in the UV

The lipid extract was dissolved in iso-octane and a 1% (*w*/*v*) and the solution was read at 232 nm, 266 nm, 268 nm and 274 nm, in a double-ray Agilent spectrophotometer model 89090A (Santa Clara, CA, USA) (Consleg, Annex IX) [5].

#### 2.4.5. Antioxidant Activity (DPPH Assay on the Extracted Fat)

The DPPH assay of the extracted fat was performed without extracting antioxidant compounds from the fat. It was conducted in an UV/Vis Spectrometer λ_2_ from Perkin Elmer (Waltham, MA, USA) using the method proposed by Kalantzakis et al. [25], modified as follows. The fat was diluted with ethyl acetate (1/10, *v*/*v*). Then, 0.25 mL of diluted oil was added to 2.25 mL of a 10^−4^ M DPPH● solution, which was previously prepared with ethyl acetate. Thus, the absorbance of the mixture was immediately measured at 515 nm (abs t0) and after 30 min of shaking and incubation in the dark (abs t30). The % of inhibition was calculated as follows: ((abs t0 − abs t30)/abs t0)) * 100. The radical scavenging activity (% of inhibition) was compared with a Trolox calibration curve and results were then expressed as TEAC values (μmol TE∙100 g^−1^ of fat extracted).

#### 2.4.6. FAMEs

An aliquot of the lipid fraction was methylated using the annex XB method A [5]. Gas chromatographic analysis of FAMEs was conducted as described in a previous work [26] and the results are expressed as % m/m.

### 2.5. Statistical Analysis

All analyses were carried out on three batches of Cantuccini biscuits, each one produced on a different day. For each batch, two replicates were conducted, each one from a different package. In brief, each result was the mean of six analyses. Means, standard deviations, R^2^ values and *r* values were obtained using Excel 2010 software. The t-test was calculated at the 95% confidence interval. Analysis of variance (one-way ANOVA) and Tukey’s tests were performed to determine the significance of differences. A *p*-value < 0.05 was set as the significance level using SPSS software version 17.0.

## 3. Results and Discussion

### 3.1. Cantuccini Biscuits

#### 3.1.1. Cantuccini Colour

A colour analysis is a fast, easy and non-destructive test that can be applied to a bakery product. If the physicochemical and hedonic properties are considered, the first parameter used by a consumer in food acceptability is colour [27,28,29]. Changes in colour (both darkening and discolouration) during the biscuits’ shelf-life influence the consumer’s evaluation at the time of purchase [30,31]. In the studied biscuits, *L** was highest in Cantuccini prepared with the original recipe; thereafter, the highest lightness value was found in the Cantuccini prepared with the modified recipe, showing a tendency to clear with time in biscuits prepared with 70% EVOO. The same initial trend but for a 4 month storage period was found for *L** inside the biscuits (Table 1).

Colour did not vary with the storage duration, except for *a** (*p* < 0.01) and *b** (*p* < 0.01) in the external measurement of the modified recipe (Table 2 and Table 3). In almost all storage times, the *a** value measured externally was higher in Cantuccini prepared with the OR than with the MR. This means that cow’s butter and margarine lead to a reddening of the biscuits’ external surface, but there is an inverse effect on the cut surface of the biscuits (internal side); in addition, on the external side of the MR Cantuccini, a decreasing tendency in *a** values was found (Table 1).

The *b** value measured externally was highest in OR Cantuccini until 4 months of storage, after which the highest values were generally found in MR Cantuccini. The *b** value measured internally was almost always highest in MR Cantuccini and varied between a low of 27.55 to a high of 29.74. The *b** value measured in MR Cantuccini showed a very high significant decrease during storage (*p* < 0.001) (Table 3). These values indicate the tendency for both OR and MR biscuits to turn more yellow in the last period of storage (Table 3). The chroma showed significant differences both during storage and between the different formulations on the internal and external sides. Chroma of the external side always showed higher values than those measured on the internal side (Table 4).

De Morais et al. [32] studied the storage stability of sweet biscuits prepared with recovered potato starch and found an initial *L** increase (36 d), a subsequent decrease until 144 d and a final increase to 180 d; in the same study, in agreement with our results, a substantial increase in the *b** value was found during the first 108 days, with a subsequent decrease up to 180 d.

#### 3.1.2. Relative Humidity (RH)

Moisture content plays a fundamental role in the manufacturing process of bakery products, including Cantuccini biscuits; furthermore, it influences their flavour, fragrance and consistency.

RH was always highest in OR Cantuccini, from 4.08 at T0 to 6.23 at T12, with a 52.7% increase in one year (*p* < 0.001), whereas RH in MR Cantuccini varied from 3.55 (T0) to 4.56 (T12), with a significant 28.5% increase (*p* < 0.001) in the same storage period (Figure 1a). It can be observed that Cantuccini prepared with the MR showed an initial lower RH than those prepared with the OR. Probably the different fat composition in OR (only butter and margarine) facilitated the absorption of moisture by the biscuits before being packaged. This process also continued after packing for two main reasons: (i) biscuits were not vacuum-packed and the package contained more or less 50% (*v*/*v*) of air included in the package at the time of sealing; (ii) a little vapour exchange also occurred during the one year of storage with the external atmosphere because the water vapour barrier of the packaging material used by the producer was high but not very high. It has to be pointed out that normally Cantuccini are consumed within 3–4 months of production, and in both the studied recipes RH showed low increases during the first six months, after which the increases were more evident (Figure 1a).

The initial RH values (T0) were confirmed by findings of Robertson [33], who found 1–5% of RH in just-baked biscuits after cooling at room temperature. Dried products are characterised by hygroscopic behaviour, which causes water absorption influenced by the atmosphere and the packaging materials. Other authors found RH increases ranging from 2% to 5.56% in a 105 d experiment with different packaging materials. They pointed out that the biscuits packed with film containing poly lactic acid were the most sensitive to RH absorption [34]. Similar behaviour was found by Chowdhury et al. [35], who over 3 months’ storage of local and foreign biscuits, found constant RH increases of between 1 and 5.34% (T0) and between 2.81 and 4.64% at the end of the study. In agreement with the Cantuccini analyses, a constant increasing aw trend was found by other authors who studied biscuits prepared with wheat flour or alternatively corn, sunflower or high-oleic sunflower oil [36].

#### 3.1.3. Water Activity (aw)

The aw showed very high and significant increasing trends in both recipes (Figure 1b). In detail, the highest aw value was found in OR Cantuccini with 10 months’ storage, while only on the last storage date was the aw higher in MR Cantuccini. Overall, in biscuits prepared with OR, aw moved from a low of 0.283 at T0 to 0.357 at T12 (*p* < 0.001), whereas in MR Cantuccini, the values moved from 0.223 (T0) to 0.390 (T12) (*p* < 0.001). Data for aw were found to be correlated with RH values as follows: R^2^ = 0.5636 and *r* = 0.7507 for OR; R^2^ = 0.8527 and *r* = 0.9234 for MR. As with RH, for aw the values also increased slightly for the first 6 months and more consistently from the 6th to the 12th month.

This is in agreement with the findings of Hough et al. [37], who found a positive correlation between RH and aw in a study on different bakery products, including toasted bread, crackers from whole-wheat flour, crackers from normal flour and vanilla-flavoured sweet biscuits.

The modified Cantuccini recipe contains reduced amounts of both the RH and aw (Figure 1a,b). This was due to the capacity of shortening to coat the gluten network and starch particles in dough with an extensible film during mixing, which reduces the water hydration capacity of the dough [38]. Margarine and cow’s butter contain 10–15% water, whereas EVOO does not contain water. This could explain the major hydrophobic behaviour of MR with respect to OR and the lower RH and aw values.

These values were lower than those found in Amaretti cookies (0.64–0.80) that were individually packaged in a high water vapour barrier film and stored for 120 days [39]. The increasing trend for aw in Cantuccini worsens the shelf-life, even if the aw values in both recipes remained always well within the range of 0.6–0.8, which is a critical factor for the growth of moulds [40].

#### 3.1.4. Cantuccini Texture

Texture is often considered by consumers at a subconscious level. The evaluation of the texture is not immediate because consumers often concentrate initially on flavour [41]; however, if the biscuit is not crisp and crunchy and is instead stale it will be rejected, even if the flavour is excellent.

Crispy is defined as, “A dry, rigid food, which when bitten with the incisors, fractures quickly, easily and totally, while emitting a relatively loud, high-pitched sound”, whereas crunchy is defined, “A dense-textured food, which when chewed with the molars, undergoes a series of fractures while emitting relatively loud, low-pitched sounds” [42].

Hardness values decreased constantly over the 12 month storage period for both fat formulations. These values varied between a low of 3659 and a high of 5619 in the MR biscuits and between 2723 and 6150 in the OR biscuits, increasing by 2.25 times in the OR (*p* < 0.001) and 1.54 times in the MR Cantuccini (*p* < 0.05) (Figure 2a). The hardness value in the control (T0) was higher in the OR, although after two months’ storage (T2) the OR values were always lower than the MR ones. This means that the recipe variation, with the addition of extra virgin olive oil, the elimination of margarine and the reduction in cow’s butter content, produced a biscuit that maintained greater hardness during its shelf-life (Figure 2a).

Hough et al. studied different bakery products and found a negative correlation between aw and texture, as well as differences in relations to the biscuit composition and the type of flour [37]. Piga et al. studied the evolution of typical Italian ‘Amaretti’ cookies during storage and found a better response in term of delays in hardening in cookies packaged with an aluminium foil with respect to cookies wrapped in polyvinylchloride. They explained that this was possibly due to water redistribution, with consequent sucrose crystallization and a consequent change to a mealy texture from a soft and moist one [43].

Fracturability value was determined as the distance taken by the probe until the point of fracture.

Fracturability values showed a non-significant decrease with storage for the OR and a significant decrease (*p* < 0.05) for the MR (Figure 2b), reflecting a lowering of the friability, which is negative for biscuits in terms of consumer acceptability.

Our results are similar to the findings by Jacob and Leelavathi [44], who studied biscuits prepared with refined wheat flour and alternative fats, including (i) emulsified bakery shortening, (ii) emulsified margarine, (iii) non-emulsified hydrogenated vegetable fat and (iv) sunflower oil, and found that cookies containing liquid oil in the recipe had a relatively harder texture compared to bakery shortening and hydrogenated fat. A similar decreasing trend in hardness and fracturability values was also described by Balestra et al. [34] in biscuits prepared with sunflower oil and stored for 120 days.

Two of the most important parameters influencing hardness and fracturability are aw and RH, which were found to be inversely correlated with hardness. These data are explained by the Pearson test and t-test. In the OR Cantuccini, the higher the aw the lower the hardness (R^2^ = 0.8498; *r* = −0.9218; t = 10.2123; *p* = 0.000), and the higher the RH the lower the hardness (R^2^ = 0.7713; *r* = −0.8782; t = 10.2020; *p* = 0.000). Similarly, in the MR Cantuccini, the higher the aw the lower the hardness (R^2^ = 0.9580; *r* = −0.9797; t = 15.9785: *p* = 0.000), and the higher RH the lower the hardness (R^2^ = 0.9092; *r* = −0.9535; t = 15.9670; *p* = 0.000).

Findings from other authors reported higher fracturability values (*p* ≤ 0.05) for formulations containing potato and corn starch with respect to biscuits prepared without eggs and without carob syrup. These results were explained by the looser matrix formation in potato and corn starch formulations, which contained lower levels of proteins [45]. Torres-González et al., in a study on Colombian lemon biscuits, described a positive and highly significant correlations (*p* < 0.01) between fracturability and firmness (*r* = 0.973), consistency (*r* = 0.933) and maximum strength (hardness) (*r* = 1), with a moderate correlation (*p* < 0.05) between fracturability and stiffness (*r* = 0.405) [46].

#### 3.1.5. Cantuccini Sensory Analysis

The sensory characteristics analysed by the panelists decreased during storage, although with different trends for OR Cantuccini and MR.

The panelists preferred the colour of OR Cantuccini for the first 8 months of storage, whereas the colour of MR Cantuccini was better evaluated from T10. At T12, the colour of the OR biscuits was considered to be of minimum acceptable quality (Figure 3a).

OR Cantuccini obtained the highest scores until T4 and were equally evaluated from T6 to T10, whereas at T12 the biscuits prepared with EVOO were preferred (Figure 3b).

The taste and flavour of OR Cantuccini were preferred in the first period of storage, although the panelists gave the same scores at T10 and T12 for taste and the same scores at T8 and T10 for flavour, with a preference for the flavour of OR at T12 (Figure 3c,d). The texture was more appreciated in OR at T0, after which the panelists preferred the biscuits prepared with the modified recipe (Figure 3e). Regarding overall acceptability, OR biscuits obtained the highest scores at T0, T2 and T8, whereas the same scores were obtained for OR and MR for the remaining sampling dates (Figure 3f).

We can speculate that the EVOO in the recipe preserved or extended the Cantuccini shelf-life and consequently the sensory properties. Our observations are in agreement with the findings of other authors, who studied the addition of phenolic extracts from olive leaves and olive mill wastewater to gluten-free breadsticks and found an amelioration in the physical–chemical–sensory properties of these bakery products [47].

Sensory properties and shelf-life were found to be related. In a study conducted on the shelf-lives of 3-month-packaged biscuits, other authors found no substantial variation in the first 2 months of storage and only slightly decreased scores for certain sensory attributes (texture, taste, flavour) after 3 months of storage [35]. 

### 3.2. Fat

#### 3.2.1. Colour of Fat

The colour values for Cantuccini fats are depicted in the Table 5. Luminance (**L***) was always highest in the OR Cantuccini, which showed a constant and significant decrease (*p* < 0.05) from 25.51 (T0) to 23.90 (T12). The same decreasing behaviour was measured in MR Cantuccini from 24.80 (T0) to 21.69 (T12) (*p* < 0.001). These values describe a browning of the fat during storage in both recipes.

The fat colour of MR Cantuccini produced the highest *a** values for the first 8 months and at the same time an increasing trend from 0.50 (T0) to 0.87 (T12) and a 74% increase in red colouring after 12 months (*p* < 0.001), whereas the *a** values in the fat of OR Cantuccini increased from 0.45 (T0) to 1.19 (T12) with an increase of 164% (*p* < 0.001). The *a** values showed constant reddening of the fat for both recipes during the shelf-life. The *b** values of OR Cantuccini moved between 3.26 (T0) and 5.08 (T12), with yellowing of the fat during the 12 month storage period, whereas the fat extracted from MR Cantuccini showed an inverse and decreasing behaviour from 5.07 (To) to 4.28 (T12) with a tendency toward blue colouring.

Chroma testing is applied to evaluate the degree of difference of a hue in comparison to a grey with the same lightness. The higher the chroma values, the higher the colour intensity of the samples as perceived by humans [24]. In both recipes, the chroma values decreased significantly during the storage year, in particular in the fat of OR Cantuccini.

Other studies have shown the influence of fat on colouring. Goubgou et al. [48] studied sorghum cookies prepared with refined cottonseed oil, refined palm oil and crude palm oil (16%, 20% and 24% in the recipe, respectively), and found browning on all cookies, with yellowing on cookies produced with red palm oil. In addition, they found that the higher the fat content in the recipe, the higher the browning intensity. In a study conducted on 3D-printed cookies, the influence of fat (butter and olive oil) on the cookies’ colour was assessed. The authors found that the colour parameter *b** and chroma values were significantly influenced by the interaction between flour and fat (*p* < 0.05), with the most intensive sample being prepared with oats and olive oil [49].

#### 3.2.2. Free Acidity

Acidity showed a very high and significant variation during storage (*p* < 0.01) in fat extracted from both OR and MR Cantuccini, with a tendency to increase with storage (Figure 4a). In detail, the acidity of OR fat was 1.61% in the control (T0) and 2.27% at T12 (47.20% increase), whereas the acidity of MR fat was 1.02% in the control (T0) and 1.35% at T12 (32.35% increase). The acidity of MR fat was lower than that of OR at all studied storage dates, in particular the difference as an absolute value in the control was 0.59%, whereas after 12 months’ storage it was 1.12%. This means that EVOO in the recipe reduced free fatty acid formation and maintained lower acidity. The lowest free acidity occurring in the fat of MR Cantuccini may have been due having to the lowest RH; in fact, moisture facilitates the hydrolytic degradation of fats.

The free acidity levels of pure fats and oils used in this study before being added to the dough were 0.22% for margarine, 0.60% for cow’s butter and 0.70% for EVOO.

Daglioglu et al. [50] studied the free acidity variation in the fats extracted using *n*-hexane from 8 different bakery products, and in all cases found significant and constant acidity increases during a 12 month storage period, with the highest value found in wafers (1.20%). Similarly, free acidity increases were found in six different biscuits prepared with six different cereals and stored with two different packaging materials and for three months [51].

#### 3.2.3. DPPH Assay

The DPPH assay is commonly applied to many matrices to test their antioxidant capacity [52,53,54].

The antioxidant activity was almost always highest in the lipid fraction of the fat obtained from the MR Cantuccini. At T0 (control), the DPPH value was 203.74 (μmol TE·100 g^−1^ fat) in fat from MR and 164.15 (μmol TE·100 g^−1^ fat) in fat from OR (i.e., 24.12% more than in the recipe containing EVOO). During 12 months’ storage, a continuous reduction was found in the antioxidant activity measured with the DPPH assay (Figure 3b). At T6, the antioxidant activity levels were 181.83 and 153.97 μmol TE·100 g^−1^ fat, respectively, for MR and OR (18.01% more than in the recipe containing EVOO). After 12 months’ storage, the antioxidant activity levels were 168.75 (MR) and 140.58 (OR) μmol TE·100 g^−1^ fat, respectively, i.e., a decrease of 20.73% from T0 to T12 for the MR and a decrease of 16.77 in the same period for the OR (Figure 4b). These data demonstrated that the antioxidant activity was positively influenced by the EVOO included in the MR during 12 months’ storage. Our findings agree with Omran et al. [55], who studied biscuits prepared with wheat flour and with different percentages of defatted and non-defatted flaxseed flour, observing a decreasing trend in the DPPH assay value over 3 months’ storage.

#### 3.2.4. Spectrophotometric Indices in the UV

The K232 value indicates the presence of dienes conjugates. The conjugated structure formed in fatty acids is important because it is more prone to oxidative reactions compared to the isolated structure and causes drying reactions in oil in the presence of oxygen. The increase in conjugate-diene formation during oxidative reactions is a very important parameter because it gives information about the state of the oil degradation.

In the studied samples, at T0 the K232 and K268 values were higher in OR fat (2.71 and 2.31, respectively) than in the MR fat (2.05 and 0.69, respectively). This was due to the high spectrophotometric K232 indices of 2.914 and 2.062, respectively, for cow’s butter and margarine as a consequence of the high conjugated diene content. At the same time, the K268 indices were also high for both cow’s butter (0.726) and margarine (0.751) in relation to the conjugated trienes.

The tendency found was an increasing trend with time for both the MR fat and OR fat. The K232 value in the OR fat was 3.44 after 12 months’ storage (26.94% more than T0) (*p* < 0.001), whereas the K232 value in the MR fat was 3.22 after 12 months’ storage (57.08% more than T0) (*p* < 0.001); this meant that the relative increase was more or less 2 times higher in the fat from the MR, although the absolute values were similar in fats from the OR and MR, especially in the last 4 months of storage (Figure 5a).

In a study conducted on Turkish margarines during 12 weeks’ storage, the K232 values were found to fluctuate in the range of 3.17–9.35 for margarines stored at 4 °C, whereas for margarines stored at 25 °C the K232 variation was found to be between 4.45 and 13.03 [56].

The K268 value in the OR fat was 3.22 after 12 months’ storage (11.69% more than T0), whereas the K268 value in the MR fat was 0.64 after 12 months’ storage, i.e., only partially different from the T0 value (Figure 5b).

Additionally, the high K268 indices were due to the fats used in OR Cantuccini, namely 0.751 for cow’s butter and 0.726 for margarine, as a consequence of the high conjugated triene content.

Schafer-De Martini Soares et al. [57] studied the K270 values in margarines prepared by enzymatic interesterification and containing palm stearin, palm kernel oil and olive oil. They found absorbance rates varying between 0.5 and 1.3 in relation to the immobilized thermostable lipases used and regardless of whether the analytical determination was conducted prior to or after the interesterification.

The ΔK value was higher in fat from OR than in fat from MR at all sampling points (0.09 and 0.02 at T0; 0.12 and 0.02 at T6; 0.21 and 0.02 at T12) (Figure 5c). It is worth noting that the ΔK value in fat from OR Cantuccini increased by 133.33% over 12 months, whereas the ΔK value calculated in fat from MR Cantuccini remained the same (0.09) over 12 months’ storage (Figure 5c). Caponio et al. [58] studied the evolution of K232 and K268 during 5 months’ storage of *Taralli* (a savoury biscuit) prepared with extra virgin olive oil, olive pomace oil and refined palm oil, and found that extra virgin olive oil always showed the lowest values for both indices. This confirms our results regarding K268, for which extra virgin olive oil showed a reduction in the formation of the secondary products of oxidation.

#### 3.2.5. FAMEs

The Trade Standard applied to olive oil by the International Olive Council [59] and the European Regulation on olive oil [5] describes the fatty acid composition of olive oil as ranging between C14:0 and C24:0. The fatty acids ranging between C4:0 and C12:0 have to be ascribed to cow’s butter [60,61]. For this reason, the C4:0 to C12:0 fatty acids were found in the highest quantities in the OR Cantuccini.

The FAME composition of the OR Cantuccini showed 36.21–38.60% of SFAs (Table 6), whereas the modified recipe contained 14.70–19.29% SFAs (Table 7). Reciprocally, the FAME composition of the original recipe showed 61.40–63.79% UFAs, whereas the modified recipe showed 80.71–85.30% UFAs. This difference was due to both the absence of margarine and the 50 to 30% reduction in cow’s butter in the modified recipe with the addition of 70% EVO. Margarine is a hydrogenated fat, and the type used in this work was found to contain 44.75% SFAs and 55.25% UFAs. Each SFA has different effects on the plasma concentrations of the various fractions of lipoproteins bound to cholesterol. For example, C12:0, C14:0 and C16:0 increase the level of cholesterol linked to high-density lipoproteins; for this reason, in recipe for bakery products, it is useful to reduce the C12:0, C14:0 and C16:0 contents [62]. Replacing SFAs (C12:0–C16:0) with MUFAs leads to reductions in LDL-bound cholesterol, as well as in the total cholesterol/HDL cholesterol ratio, and improves the insulin sensitivity [62]. The total levels of C12:0, C14:0 and C16:0 varied from 28.87–32.18% (OR) to 10.57–11.74% (MR), with reductions ranging from 2.73- to 2.74-fold. The use of olive oil in the Cantuccini recipe increased the C18:1 content from 37.55–40.65% (OR) to 59.02–63.18% (MR), i.e., from 1.57 to 1.55 times more, with an evident health benefit for the consumer. Oleic acid (C18:1), the most common MUFA in olive oil, was found to be an antiapoptotic and anti-inflammatory agent via regulation of cyclooxygenase-2 (COX-2) and inducible nitric oxide synthase (iNOS) through the activation of nuclear factor-kappa B (NF-κB), resulting in the activation of downstream inflammatory mediators [63]. High significant differences were found between the fat contents in the two recipes during Cantuccini storage. The Σ SFA values varied between 38.60% (T0) and 36.99% (T12) in OR and between 19.29% (T0) and 15.84% (T12) in MR, with the minimum content being observed at T4 (14.70%). The Σ MUFA values varied from 39.16% (T0) to 42.81% (T12) in OR and from 62.35% (T0) and 64.71% (T12) in MR, with the maximum content being observed at T4 (66.53%). C18:1 accounted for 37.55% at T0 and 40.65% at T12 in the OR, whereas it accounted for 59.02% at T0 and 61.12% at T12 in the MR. In both recipes, both the UFA/SFA and the 18:2ω6/18:3ω3 ratios increased during storage.

The changes in fatty acid composition were related to alterations during storage, which were found to be due to the initial fat content, initial water content and storage duration [64]. At the same time, the number and position of double bonds and the storage conditions affect the rate of oxidation [65,66].

Culetu et al. [67] found different fatty acid behaviour during storage in oat-based gluten-free cookies prepared with different fats, such as butter, margarine, lard, palm oil refined palm oil and hydrogenated palm oil. They found a decreasing trend for SFAs for lard and in refined palm oil in stearin cookies, consistent with our SFA results, while at the same time they found a decreasing trend for MUFAs, with the exception of the hydrogenated palm oil cookies, in substantial agreement with our findings.

## 4. Conclusions

The complete replacement of margarine and the partial replacement of butter with extra virgin olive oil in the original Cantuccini biscuit recipe led to improvements in water activity, relative humidity, hardness, K270 values, DPPH assay results and acidity, whereas worse results were found with regard to K232 and ΔK values. This means that the use of extra virgin olive oil in the recipe, in many cases studied in this work, improved the physical and chemical properties of biscuits but worsened the parameters related to the secondary products of oxidation (K270). The Cantuccini biscuit colour was instrumentally evaluated and found to be strictly related to both fats and EVOO and to the length of storage. The fatty acid methyl ester compositions varied dramatically and the use of EVOO increased the unsaturated fatty acid content, in particular monounsaturated fatty acids, with oleic acid being the prevailing compound. The sensory analysis showed an expected worsening of the biscuit quality but good acceptability of Cantuccini prepared with 70% EVOO. Extra virgin olive oil can be applied to prepare functional products that maintain the physical–chemical–sensory properties of Cantuccini. 

## Figures and Tables

**Figure 1 foods-11-00299-f001:**
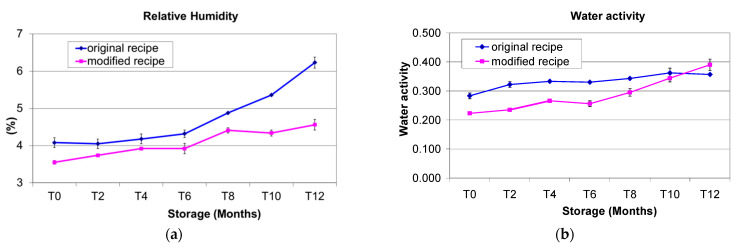
(**a**) Relative humidity of Italian Cantuccini biscuits prepared with the original recipe and the modified recipe during one year’s shelf-life. (**b**) Water activity of Italian Cantuccini biscuits prepared with the original recipe and the modified recipe during one year’s shelf-life.

**Figure 2 foods-11-00299-f002:**
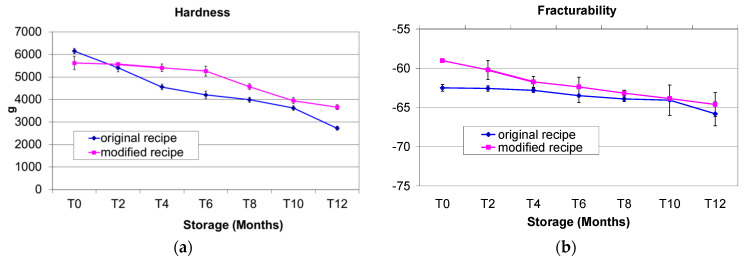
(**a**) Hardness of Italian Cantuccini biscuits prepared with the original recipe and the modified recipe during one year’s shelf-life. (**b**) Fracturability of Italian Cantuccini biscuits prepared with the original recipe and the modified recipe during one year’s shelf-life.

**Figure 3 foods-11-00299-f003:**
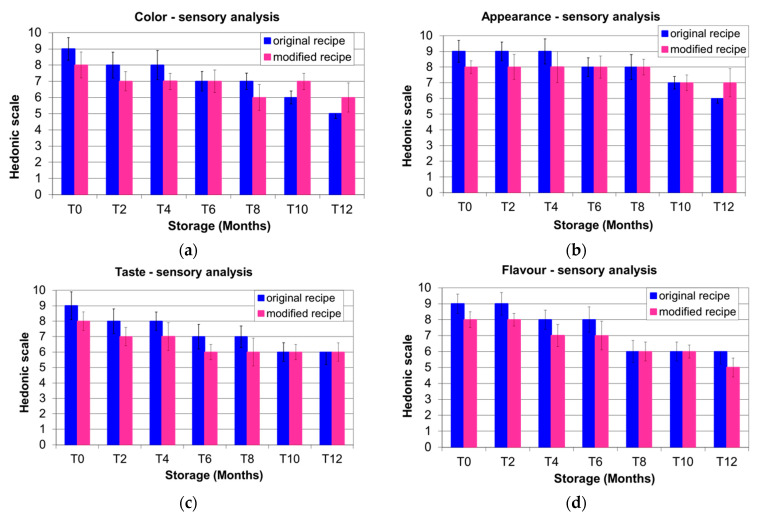
Sensory analysis of Italian Cantuccini biscuits prepared with the original recipe and the modified recipe during one year’s shelf-life: (**a**) colour; (**b**) appearance; (**c**) taste; (**d**) flavour; (**e**) texture; (**f**) overall acceptability.

**Figure 4 foods-11-00299-f004:**
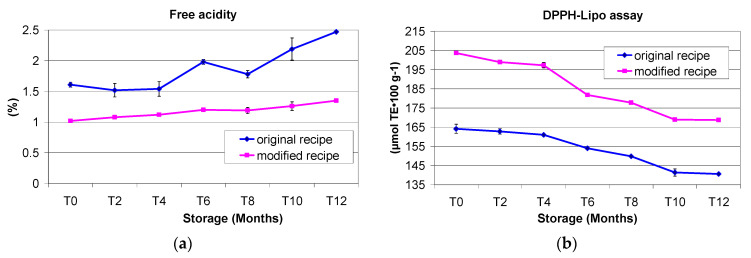
(**a**) Free acidity levels of fat samples extracted from Italian Cantuccini biscuits prepared with the original recipe and the modified recipe during one year’s shelf-life. (**b**) DPPH-Lipo assay results for fat samples extracted from Italian Cantuccini biscuits prepared with the original recipe and the modified recipe during one year’s shelf-life.

**Figure 5 foods-11-00299-f005:**
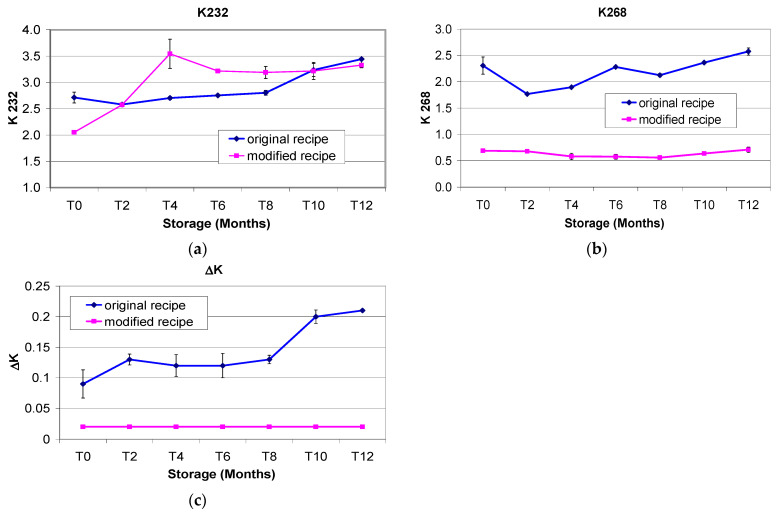
(**a**) K232 values for fat samples extracted from Italian Cantuccini biscuits prepared from the original recipe and the modified recipe during one year’s shelf-life. (**b**) K268 values for fat samples extracted from Italian Cantuccini biscuits prepared from the original recipe and the modified recipe during one year’s shelf-life; (**c**) The ΔK values for fat samples extracted from Italian Cantuccini biscuits prepared from the original recipe and the modified recipe during one year’s shelf-life.

**Table 1 foods-11-00299-t001:** Colours and **L*** values of Cantuccini biscuits made using the original recipe (OR; 50% butter/50% margarine) and modified recipe (MR; 70% EVOO/30% butter). One-way ANOVA, means ± SD (of six replicates) in the same row followed by a different letter are significantly different according to Tukey’s test; *** *p* < 0.001. Means in the same column are distinguished by capital letters. Means in the same line are distinguished by small letters.

	T0	T2	T4	T6	T8	T10	T12	*Sign.*
*L** external OR	56.59 ± 2.37 aB	56.71 ± 0.61 aB	51.65 ± 6.04 aC	49.46 ± 6.97 aB	52.7 ± 0.51 aB	57.35 ± 1.48 aB	51.13 ± 3.72 aB	n.s.
*L** external MR	51.87 ± 1.02 bB	53.37 ± 0.16 bB	55.03 ± 2.22 bB	58.7 ± 3.95 bB	55.04 ± 1.41 bB	65.70 ± 0.36 aA	67.41 ± 0.53 aA	n.s.
*L** internal OR	68.75 ± 4.96 aA	66.73 ± 5.38 aA	65.36 ± 2.21 aA	69.25 ± 1.87 aA	64.54 ± 2.28 aA	60.65 ± 0.77 aB	67.72 ± 2.91 aA	n.s.
*L** internal MR	66.03 ± 1.29 aA	55.88 ± 0.30 bB	56.11 ± 8.46 bB	69.94 ± 0.65 aA	62.6 ± 2.58 bA	66.72 ± 2.86 aA	66.99 ± 0.12 aA	***
*Sign.*	***	***	***	***	***	***	***	

**Table 2 foods-11-00299-t002:** Colours and **a* values of Cantuccini biscuits made using the original recipe (OR; 50% cow’s butter/50% margarine) and modified recipe (MR; 70% EVOO/30% cow’s butter). One-way ANOVA, means ± SD (of six replicates) in the same row followed by a different letter are significantly different according to Tukey’s test; *** *p* < 0.001. Means in the same column are distinguished by capital letters. Means in the same line are distinguished by small letters.

	T0	T2	T4	T6	T8	T10	T12	*Sign.*
*a** external OR	13.46 ± 0.23 aA	12.96 ± 0.25 aA	13.82 ± 0.82 aA	12.61 ± 1.22 aA	14.63 ± 0.37 aA	12.63 ± 1.12 aA	13.19 ± 0.65 aA	n.s.
*a** external MR	13.86 ± 1.27 aA	12.82 ± 0.28 abA	12.00 ± 0.10 abcA	13.65 ± 0.77 abA	13.13 ± 1.93 abA	11.08 ± 0.57 bcA	10.45 ± 0.12 cB	**
*a** internal OR	4.32 ± 1.06 aB	4.56 ± 0.46 aC	5.76 ± 0.15 aB	4.96 ± 1.27 aB	5.48 ± 0.82 aB	6.69 ± 1.79 aB	4.96 ± 0.91 aC	n.s.
*a** internal MR	5.8 ± 1.23 aB	6.90 ± 0.60 aB	6.88 ± 2.63 aB	5.43 ± 0.16 aB	6.01 ± 0.72 aB	5.48 ± 0.48 aB	5.70 ± 0.04 aC	n.s.
*Sign.*	***	***	***	***	***	***	***	

**Table 3 foods-11-00299-t003:** Colours and *b** values of Cantuccini biscuits made using the original recipe (OR; 50% cow’s butter/50% margarine) and modified recipe (MR; 70% EVOO/30% cow’s butter). One-way ANOVA, means ±SD (of six replicates) in the same row followed by a different letter are significantly different according to Tukey’s test; n.s., not significant; ** *p* < 0.01; *** *p* < 0.001. Means in the same column are distinguished by capital letters. Means in the same line are distinguished by small letters.

	T0	T2	T4	T6	T8	T10	T12	*Sign.*
*b** external OR	29.66 ± 0.97 aA	30.24 ± 0.40 aA	29.48 ± 1.95 aA	27.48 ± 3.25 aA	31.93 ± 0.65 aA	31.58 ± 2.49 aAB	27.8 ± 3.26 aB	n.s.
*b** external MR	29.07 ± 1.67 cA	29.80 ± 0.30 cA	29.01 ± 1.50 cA	30.13 ± 1.30 cA	31.82 ± 2.13 bcA	34.26 ± 0.67 abA	35.96 ± 0.17 aA	***
*b** internal OR	23.03 ± 1.21 aB	23.64 ± 1.75 aC	26.99 ± 2.01 aA	24.96 ± 3.38 aA	25.18 ± 1.68 aB	26.02 ± 0.88 aC	25.83 ± 1.58 aB	n.s.
*b** internal MR	28.54 ± 1.82 aA	26.67 ± 0.54 aB	26.35 ± 1.11 aA	28.70 ± 0.53 aA	29.74 ± 0.68 aA	28.63 ± 2.03 aBC	28.85 ± 0.11 aB	*
*Sign.*	**	***	n.s.	n.s.	***	**	***	

**Table 4 foods-11-00299-t004:** Colours and chroma values of Cantuccini biscuits made using the original recipe (OR; 50% cow’s butter/50% margarine) and modified recipe (MR; 70% EVOO/30% cow’s butter). One-way ANOVA, means ±SD (of six replicates) in the same row followed by a different letter are significantly different according to Tukey’s test; * *p* < 0.05; ** *p* < 0.01; *** *p* < 0.001. Means in the same column are distinguished by capital letters. Means in the same line are distinguished by small letters.

	T0	T2	T4	T6	T8	T10	T12	*Sign.*
Chroma external OR	32.57 bA	32.90 bA	32.56 bA	30.24 cB	35.12 aA	34.01 aB	30.77 cB	*
Chroma external MR	32.21 dA	32.44 dA	31.39 eA	33.08 cA	34.42 cAB	36.01 bA	37.45 aA	**
Chroma internal OR	23.43 dC	24.08 dC	27.60 aB	25.45 cC	25.77 cD	26.87 bD	26.30 bD	*
Chroma internal MR	29.12 bB	27.55 cB	27.23 cB	29.21 bB	30.34 aC	29.15 bC	29.41 bBC	*
*Sign.*	**	**	*	**	***	***	***	

**Table 5 foods-11-00299-t005:** Colour of fat samples from Cantuccini biscuits made from the original recipe (OR; 50% cow’s butter/50% margarine) and modified recipe (MR; 70% EVOO/30% cow’s butter). One-way ANOVA, means ± SD (of six replicates) in the same row followed by a different letter are significantly different according to Tukey’s test; * *p* < 0.05; *** *p* < 0.001.

	T0	T2	T4	T6	T8	T10	T12	*Sign.*
*L** OR	25.51 a	25.04 a	25.44 a	24.79 a	24.88 a	24.17 b	23.90 b	*
*L** MR	24.80 a	24.24 a	23.44 a	22.80 b	22.32 b	21.66 b	21.69 b	***
*a** OR	0.45 d	0.50 d	0.60 c	0.65 c	0.66 c	0.94 b	1.19 a	***
*a** MR	0.50 d	0.54 d	0.71 c	0.76 bc	0.76 bc	0.80 ab	0.87 a	***
*b** OR	5.08 a	4.95 b	3.90 c	3.95 c	3.64 d	3.39 e	3.26 e	***
*b** MR	5.07 a	4.89 b	4.62 c	4.65 c	4.55 d	4.31 e	4.28 e	***
Chroma OR	5.10 a	4.98 ab	3.95 c	4.00 c	3.70 d	3.52 d	3.47 e	***
Chroma MR	5.09 a	4.92 b	4.67 c	4.71 c	4.61 cd	4.38 e	4.37 e	***

**Table 6 foods-11-00299-t006:** Fatty acid methyl esters found in the original recipe Cantuccini. One-way ANOVA, means ± SD (of six replicates) in the same column followed by a different letter are significantly different according to Tukey’s test; *** *p* < 0.001; ** *p* < 0.01; * *p* < 0.05; n.s., not significant.

(a)																			
	C4:0	C6:0	C8:0	C10:0	C12:0	C14:0	C16:0	C16:1	C17:0	C17:1	C18:0	C18:1	C18:1w7	C18:2	C18:2TT	C18:2CT	C18:2TC	C18:3w3	C18:3w6
**T0**	1.54 e	0.17 a	0.01 a	0.01 a	0.14 a	0.70 a	31.34 a	0.43 ab	0.09 ab	0.04 b	4.12 a	37.55 d	0.90 b	21.27 a	0.21 b	0.19 b	0.01 a	0.53 a	0.03 a
**T2**	1.93 c	0.11 bc	0.01 a	0.01 a	0.11 b	0.60 b	28.98 bc	0.44 ab	0.09 abc	0.04 ab	3.90 a	40.12 ab	1.13 ab	20.96 a	0.19 b	0.18 bc	tr	0.47 b	0.01 a
**T4**	3.15 a	0.12 b	0.01 a	0.01 a	0.10 b	0.58 b	28.19 c	0.41 b	0.10 a	0.04 b	5.10 a	40.07 b	1.42 a	19.18 d	0.18 b	0.15 c	tr	0.43 bc	0.02 a
**T6**	2.45 b	0.13 b	0.01 a	0.01 a	0.11 b	0.61 b	30.49 ab	0.41 b	0.10 a	0.04 ab	3.89 a	38.66 c	1.18 ab	20.34 abc	0.20 b	0.17 bc	0.01 a	0.45 b	0.03 a
**T8**	1.69 d	0.12 b	0.01 a	0.01 a	0.11 b	0.61 b	31.11 a	0.43 ab	0.08 c	0.04 b	3.95 a	38.89 c	1.07 ab	20.41 ab	0.21 b	0.15 c	tr	0.42 bc	0.02 a
**T10**	0.62 f	0.09 cd	0.01 a	0.01 a	0.11 b	0.61 b	31.28 a	0.44 ab	0.08 bc	0.04 b	3.86 a	40.56 ab	1.38 a	19.42 bcd	0.27 a	0.20 ab	tr	0.40 c	0.01 a
**T12**	0.59 f	0.08 d	0.01 a	0.01 a	0.10 b	0.58 b	31.36 a	0.46 a	0.09 ab	0.05 a	3.76 a	40.65 a	1.46 a	19.32 cd	0.26 a	0.22 a	tr	0.39 c	0.01 a
** *Sign.* **	***	***	n.s.	n.s.	***	***	***	**	***	**	n.s.	***	**	***	***	***	***	***	*
**(b)**																			
	**C20:0**	**C20:1**	**C22:0**	**C22:1**	**C24:0**	**ΣSFAs**		**ΣUFAs**		**ΣMUFAs**		**ΣPUFAs**		**UFAs/SFAs**		**18:2ω6/18:3ω3**	**C12:0 + C14:0 + C16:0**		
**T0**	0.27 a	0.20 a	0.13 a	0.04 ab	0.08 a	38.60		61.40 d		39.16 c		22.25 a		1.59 c		40.13 c	32.18 a		
**T2**	0.26 ab	0.20 a	0.13 a	0.06 a	0.08 a	36.21 d		63.79 a		41.99 a		21.80 a		1.76 a		44.93 b	29.69 d		
**T4**	0.27 a	0.19 bc	0.14 a	0.06 a	0.07 a	37.85 b		62.15 c		42.19 a		19.96 c		1.64 c		45.12 b	28.87 e		
**T6**	0.27 a	0.19 ab	0.12 a	0.06 a	0.08 a	38.26 ab		61.74 cd		40.54 b		21.20 ab		1.61 c		45.21 b	31.21 c		
**T8**	0.27 a	0.18 cd	0.12 a	0.04 ab	0.07 a	38.15 ab		61.85 cd		40.65 b		21.20 ab		1.62 c		49.02 ab	31.83 b		
**T10**	0.24 b	0.17 d	0.11 a	0.02 b	0.07 a	37.09 c		62.91 b		42.61 a		20.30 bc		1.70 b		49.01 ab	31.99 b		
**T12**	0.22 c	0.15 e	0.10 a	0.02 b	0.08 a	36.99 c		63.01 b		42.81 a		20.20 bc		1.70 b		49.96 a	32.04 a		
** *Sign.* **	***	***	n.s.	n.s.	***	***		***		**		***		**		n.s.	***		

**Table 7 foods-11-00299-t007:** Fatty acid methyl esters found in the modified recipe Cantuccini. One-way ANOVA, means ± SD (of six replicates) in the same column followed by a different letter are significantly different according to Tukey’s test; *** *p* < 0.001; n.s., not significant.

(a)																			
	C4:0	C6:0	C8:0	C10:0	C12:0	C14:0	C16:0	C16:1	C17:0	C17:1	C18:0	C18:1	C18:1w7	C18:2	C18:2TT	C18:2CT	C18:2TC	C18:3w3	C18:3w6
**T0**	2.92 a	0.02 a	0.07 b	0.07 b	0.45 a	0.30 a	10.99 a	0.94 a	0.11 ab	0.17 a	3.88 a	59.02 e	1.95 b	17.89 d	0.03	0.01	0.01	0.42	tr
**T2**	2.46 b	0.01 b	0.65 a	0.65 a	0.30 e	0.29 a	10.40 b	0.89 bc	0.12 a	0.18 a	3.58 b	59.34 e	2.03 b	17.96 cd	0.02	0.01	0.01	0.40	tr
**T4**	0.74 e	0.01 b	0.04 b	0.06 c	0.35 bc	0.25 bc	10.41 b	0.86 c	0.11 ab	0.17 a	2.31 d	63.18 a	2.06 b	18.43 b	tr	tr	tr	0.34	tr
**T6**	1.03 d	0.01 b	0.07 b	0.06 b	0.36 b	0.24 c	9.97 c	0.82 d	0.08 c	0.15 c	3.32 c	62.32 b	2.31 ab	18.20 bc	tr	tr	tr	0.36	tr
**T8**	0.78 e	0.01 b	0.05 b	0.05 c	0.34 c	0.28 ab	10.74 a	0.90 b	0.11 ab	0.18 a	3.67 b	61.61 c	2.54 a	17.55 e	tr	tr	tr	0.40	tr
**T10**	1.42 c	0.01 b	0.05 b	0.05 c	0.35 bc	0.25 b	9.97 c	0.81 d	0.10 bc	0.17 ab	3.23 c	61.11 d	2.22 ab	19.12 a	tr	tr	tr	0.35	tr
**T12**	1.42 c	0.01 b	0.05 b	0.06 c	0.35 b	0.24 bc	10.00 c	0.83 d	0.11 ab	0.16 bc	3.25 c	61.12 d	2.23 ab	19.12 a	tr	tr	tr	0.33	tr
** *Sign.* **	***	***	***	***	***	***	***	***	***	***	***	***	***	***	n.s.	n.s.	n.s.	***	n.s.
**(b)**																			
	**C20:0**	**C20:1**	**C22:0**	**C22:1**	**C24:0**	**ΣSFAs**	**ΣUFAs**		**ΣMUFAs**		**ΣPUFAs**		**UFAs/SFAs**		**18:2ω6/18:3ω3**		**C12:0+C14:0+C16:0**		
**T0**	0.29 a	0.21 a	0.13 a	0.05 d	0.07 a	19.29 a	80.71 e		62.35 f		18.36 c		4.18 d		42.60 d		11.74 a		
**T2**	0.27 abc	0.19 ab	0.11 bcd	0.07 cd	0.07 a	18.91 b	81.09 d		62.69 e		18.40 c		4.29 d		45.29 d		11.00 ab		
**T4**	0.27 abc	0.17 c	0.10 cd	0.08 c	0.05 bc	14.70 e	85.30 a		66.53 a		18.77 b		5.80 a		54.21 b		11.00 ab		
**T6**	0.26 bc	0.17 c	0.09 d	0.12 b	0.05 c	15.54 d	84.46 b		65.90 b		18.56 bc		5.43 b		50.14 c		10.57 c		
**T8**	0.29 ab	0.19 ab	0.12 ab	0.11 b	0.06 ab	16.51 c	83.49 e		65.54 c		17.96 d		5.06 c		43.54 d		11.36 a		
**T10**	0.25 c	0.20 ab	0.10 cd	0.18 a	0.06 abc	15.84 d	84.16 b		64.69 d		19.47 a		5.31 b		55.16 b		10.57 c		
**T12**	0.26 bc	0.19 bc	0.12 abc	0.19 a	0.05 bc	15.84 d	84.16 b		64.71 d		19.45 a		5.31 b		58.54 a		10.57 c		
***Sign*.**	***	***	***	***	***	***	***		***		***		***		***		***		

## Data Availability

Not available.

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
