# Peer review of "Effects of Shortening Replacement with Extra Virgin Olive Oil on the Physical–Chemical–Sensory Properties of Italian Cantuccini Biscuits"

_foods, 2022, doi:10.3390/foods11030299_

Round 1

Reviewer 1 Report

The paper titled “Effect of shortening replacement with extra virgin olive oil on physico-chemical properties of Italian Cantuccini biscuits” investigated the shelf-life and the physicochemical properties of Cantuccini biscuits with their original recipe containing only cow butter and margarine, and the effect of partial shortening replacement with extra virgin olive oil over one-year’s storage. The results revealed that the use of extra virgin olive oil in the recipe improved the physical and chemical properties of biscuits, and increased the unsaturated fatty acid content. However, the paper is lacking of enough innovation, and enough discussions and explanations of experimental data. Moreover, the discussion towards some important phenomena or mechanisms also needs to be presented, and the sentences should be clarified and improved. 

Other comments are shown as follows:

  1. Why choose the modified biscuit recipe with 30% cow butter and 70% extra virgin olive oil? The effects of different olive oil contents to replace the shortening in biscuits should be further studied.

  1. The effect of partial shortening replacement with extra virgin olive oil on the microstructure of Cantuccini biscuits should be investigated. Moreover, to compare the changes in colour of studied biscuits, the appearance of biscuits should be presented.

  1. The effects of partial shortening replacement with extra virgin olive oil on the physicochemical properties and structure of baked doughs should be studied.

  1. Why the modified recipe with extra virgin olive oil could reduce the relative humidity or water activity, and reduce free fatty acid formation?

  1. The discussion towards some important phenomena or mechanisms also needed to be presented.

  1. The authors should check the manuscript carefully for the spelling and grammatical errors. The sentence tense for the manuscript needed to be corrected carefully.

Author Response

Response to Reviewer 1

RESPONSE: Thank you very much for your suggestions and changes requested, which have undoubtedly improved our paper. You will find below responses to your questions.

  1. Why choose the modified biscuit recipe with 30% cow butter and 70% extra virgin olive oil? The effects of different olive oil contents to replace the shortening in biscuits should be further studied.

RESPONSE: 2.1 section

This choice in the modified recipe was applied after tests on the consistency and pleasantness of the Cantuccini. This ratio was applied to do not distort the type of product and to have the right compromise between tradition and innovation.

  1. The effect of partial shortening replacement with extra virgin olive oil on the microstructure of Cantuccini biscuits should be investigated. Moreover, to compare the changes in colour of studied biscuits, the appearance of biscuits should be presented.

 RESPONSE: Thank you. The investigation of the miscrostructure of Cantuccini biscuits was not investigated in this study but it is an improtant suggestion for our next work. The change of colour was described both instrumentally (Lab detection with the Minolta instrument) and now we have also included the sensory analysis of colour. 

  1. The effects of partial shortening replacement with extra virgin olive oil on the physicochemical properties and structure of baked doughs should be studied.

RESPONSE: Thank you, this is an interesting suggestion of the Reviewer for next study.

  1. Why the modified recipe with extra virgin olive oil could reduce the relative humidity or water activity, and reduce free fatty acid formation?

RESPONSE: thank you for your suggestions. Answares are included in the 3.1.2. and in the 3.2.2. sub-sections.

  1. The discussion towards some important phenomena or mechanisms also needed to be presented.

 RESPONSE: the discussion was improved, please, see the blue and red parts of the manuscript.

.

  1. The authors should check the manuscript carefully for the spelling and grammatical errors. The sentence tense for the manuscript needed to be corrected carefully.

RESPONSE: the manuscript was reviewed by an English teacher of English language. We have attached the declaration.

For every communication sent to:

Angelo Maria Giuffrè

Dipartimento AGRARIA

Università degli Studi Mediterranea di Reggio Calabria.

89124 Contrada Melissari– Reggio Calabria (Italy)

With best regards

Angelo Maria Giuffrè

Reviewer 2 Report

This paper deals with Effect of shortening replacement with extra virgin olive oil on physico-chemical properties of Italian Cantuccini biscuits.

Title is interesting and the manuscript has written and organized well but there are some issues needs more attention.

Introduction have to revise and focus on gape of research and importance of current study.

Explain about logic of used recipe containing 11% fat  in the ratio 30% cow butter and 70% extra virgin olive oil.

It was necessary to report physico-chemical property and changes of products during shelf-life.

Sensory evaluation of products was necessary but have not reported.

Hardness and fracturability results needs to describe more and discuss.

Findings about Colour of fat needs discuss more and interpret comprehensively.

Discussion about fatty acids profile and comparison between two formula is necessary.

Conclusion is too long and repetitive, it is recommended to rewrite this section after revising results and discussion.

Author Response

Response to Reviewer 2

Comments and Suggestions for Authors

RESPONSE: Thank you very much for your suggestions and changes requested, which have undoubtedly improved our paper. You will find below responses to your questions.

This paper deals with Effect of shortening replacement with extra virgin olive oil on physico-chemical properties of Italian Cantuccini biscuits.

Title is interesting and the manuscript has written and organized well but there are some issues needs more attention.

1.Introduction have to revise and focus on gape of research and importance of current study.

RESPONSE: the aim of this study was better describesd.

2.Explain about logic of used recipe containing 11% fat  in the ratio 30% cow butter and 70% extra virgin olive oil.

RESPONSE: This choice in the modified recipe was applied after tests on the consistency and pleasantness of the Cantuccini. This ratio was applied to do not distort the type of product and to have the right compromise between tradition and innovation. The shortening reduction has led to the need for an increase in the EVOO to maintain the original characteristics of the biscuits as much as possible.

3.It was necessary to report physico-chemical property and changes of products during shelf-life.

RESPONSE: physico-chemical properties and changes of products during shelf-life were better discussed.

4.Sensory evaluation of products was necessary but have not reported.

RESPONSE: sensory evaluation data were selected to prepare another manuscript, but as per suggestion of Reviewer are now included here.

5.Hardness and fracturability results needs to describe more and discuss.

RESPONSE: description and discussion about hardness and fracturability were improved;

6.Findings about Colour of fat needs discuss more and interpret comprehensively.

RESPONSE: Findings about Colour of fat are now better discussed.

7.Discussion about fatty acids profile and comparison between two formula is necessary. RESPONSE: Comparison between two formula of fatty acids was conducted.

8.Conclusion is too long and repetitive, it is recommended to rewrite this section after revising results and discussion.

Conclusion was revised in light of the Reviewer suggestion.

For every communication sent to:

Angelo Maria Giuffrè

Dipartimento AGRARIA

Università degli Studi Mediterranea di Reggio Calabria.

89124 Contrada Melissari– Reggio Calabria (Italy)

With best regards

Angelo Maria Giuffrè

Round 2

Reviewer 1 Report

the manuscript has been well revised.